



# Brief communication: Combining borehole temperature, borehole piezometer and cross-borehole electrical resistivity tomography measurements to investigate changes in ice-rich mountain permafrost

Marcia Phillips[1,2], Chasper Buchli[1], Samuel Weber[1,2], Jacopo Boaga[3], Mirko Pavoni[3], Alexander Bast[1,2]

[1]WSL Institute for Snow and Avalanche Research SLF, Flüelastrasse 11, 7260 Davos Dorf, Switzerland
[2]Climate Change, Extremes and Natural Hazards in Alpine Regions Research Center CERC, Flüelastrasse 11, 7260 Davos Dorf, Switzerland
[3]Department of Geosciences, University of Padova, Via Gradenigo 6, 35131 Padova, Italy


*Correspondence to*: Marcia Phillips (phillips@slf.ch)

**Abstract**

A novel combination of borehole temperature, borehole piezometer and cross-borehole electrical resistivity tomography (ERT) data are used to investigate changing ice-/water contents in the creeping ice-rich Schafberg rock glacier in the Eastern Swiss
Alps. Instrumentation techniques and first results are presented. The rock glacier is close to its melting point and has locally heterogeneous stratigraphies, ice-/water contents and temperature regimes. The measurement techniques presented continuously monitor temporal and spatial phase changes to a depth of 12 m and should provide the basis for a better understanding of accelerating rock glacier kinematics and future water availability.

## 1 Introduction

A widespread acceleration of ice-rich rock glaciers is being recorded in the Alps (Permos, 2019), increasing the likelihood of mass movements such as debris flows from their frontal lobes in steep terrain connected to streams (Kummert and Delaloye, 2018). The acceleration has been attributed to ice warming induced by global warming and to rising water contents within the ice-rich permafrost (Cicoira et al., 2019). Ice and water contents of rock glaciers have been modelled to investigate future water availability (Pruessner et al., 2022), and the water-related dynamics of rock glaciers have been investigated based on
aerial photography, in-situ GNSS and meteorological data (Wirz et al., 2016), as well as snow cover timing (Kenner et al., 2019). Rock glacier outflow has been quantified (Krainer and Mostler, 2002), and dynamic liquid water storage capacities estimated (Wagner et al., 2021). However, there is little direct information on the internal hydrology of rock glaciers (Zenklusen Mutter and Phillips, 2012), changes in ice-/water contents or talik formation. A substantial unfrozen water content can persist well below 0°C, depending on soil properties, salinity and pore water pressure (Arenson et al., 2022). Many Alpine
rock glaciers are close to their melting point (PERMOS 2019), and borehole temperature data do not allow to distinguish between ice and water close to 0°C, so relative changes in ice-/water content have to be monitored using geophysical methods detecting changes in resistivity (Mollaret et al., 2019). Continuous surface electrical resistivity data (ERT) provide valuable 2D information on resistivity changes in ice-rich permafrost substrates. However, 2D ERT soundings can only provide limited information with increasing depth. Cross-borehole ERT overcomes this limitation by using at least two vertical boreholes for
electrode locations (Binley, 2015). Until present, the method has primarily focused on groundwater research, especially on the remediation of contaminated groundwater (Binley and Slater, 2020) or to determine substrate characteristics, but has never been applied in mountain permafrost environments.

Time domain reflectometry (TDR) measurements in the active layer of mountain permafrost (Rist and Phillips, 2005) revealed how water contents vary on a daily and seasonal basis in frozen talus slopes. Laboratory experiments (Harris and Davies, 1998)
gave first indications of the relation between pore-water pressure and temperature during phase change in permafrost. Although



pore-water pressure is routinely monitored in other environments, piezometer data have not been collected in ice-rich mountain permafrost. Here we present a novel combination of borehole temperature, borehole piezometer and cross-borehole ERT measurements designed to investigate the changes occurring in an ice-rich rock glacier, in particular modifications of ice-to-water ratio near 0°C, with continuous measurements. Previously collected borehole temperature data, surface ERT, seismics

and electromagnetic frequency domain data indicated the occurrence of heterogeneously distributed ground ice and possible talik formation (Boaga et al., 2020) at the Schafberg rock glacier study site we focus on here.

## 2 Site and Methods

### 2.1 Borehole drilling

The monitoring site (46°29'50.391" N, 9°55'34.779" E) is on the ice-rich Schafberg rock glacier, at 2'750 m asl above Pontresina (Engadin, Eastern Swiss Alps, Fig. 1a), where an existing PERMOS borehole (Schafberg B1, drilled in 1990) is equipped with temperature sensors (Vonder Mühll and Holub, 1992). Additionally, three vertical boreholes (B3, 12 m deep; B4, 13 m; and B5, 9 m) were drilled destructively in August 2020 and stratigraphy was recorded on the basis of drill performance and observation of material ejected by air-flushing (Fig. 1b). A 4 m long PVC pipe was temporarily installed

during drilling to prevent active layer collapse (Fig. 2a). The sensors were lowered into the boreholes immediately after drill extraction, the stabilizing PVC pipe removed and the boreholes filled with a sand/gravel mixture (total volume 0.3 m³, consisting of 50% sand (≤2 mm diameter) and of 50% gravel (2-4 mm diameter)) to establish contact between the sensors and borehole walls (see sections 2.2 and 2.3) and to minimize air circulation. At the ground surface, the boreholes and instrument boxes (Fig. 2d) are protected by concrete chambers with iron lids.

### 2.2 Borehole temperatures and piezometer sensors


Borehole B1 is equipped with 16 *YSI* thermistors (type 44006) to a depth of 15.9 m (depths to 9.2 m shown in Fig. 1b), with an accuracy of ± 0.1°C and with a *Campbell* CR1000 data logger. Energy is supplied by two 12V batteries. Data are recorded every two hours and read out using a Toughbook during site visits. The borehole is part of the PERMOS network and the dataset is available online ([www.permos.ch](www.permos.ch), [doi:10.13093/permos-2021-01.](doi:10.13093/permos-2021-01.)).

Borehole B5 is located 10 m Northwest of B3 (Fig. 1c) and was equipped with ten *Keller PAA-36XiW* piezometers. The piezometer data provide temporal evolution of the effective pressure measured at the sensor's membrane (measured relative to a vacuum; pressure range 60-230 kPa, accuracy ± 11.5 kPa), combined with ten PT 1000 temperature sensors (accuracy ± 0.1°C) between 2.0 and 8.5 m depth (Fig. 1b & 1c). For protection, the sensors were smeared with Vaseline and wrapped in a thin textile (Fig. 2b). They are connected to two *Keller ARC-1 Box 4G* data loggers containing a barometer and the data are

measured hourly and transmitted daily via a mobile phone network to a cloud-based data platform.

### 2.3 Cross-borehole electrical resistivity tomography (ERT)

Boreholes B3 and B4 are 5 m apart (Fig. 1c) and are both equipped with an ERT multi-core cable (24 electrodes per borehole with 50 cm vertical spacing, at 0-11.5 m depth, see Fig. 1b). In addition, stainless steel rings were mounted at the cables' take-outs to improve electrode contact with the substrate (Fig. 2c). We use a *Syscal* system ([www.iris-instruments.com](www.iris-instruments.com)) to

automatically collect and transmit the data consisting of the *Syscal Pro Switch* 48 resistivity meter and the *Syscal Monitoring Unit SMU*, which are protected by a concrete chamber (Fig. 2d) at the ground surface between the two boreholes. The *SMU* starts, logs and transmits the data daily. The system requires two 12 V lead crystal batteries (100 Ah for current transmission, and 55 Ah for current reception), which are fed by two solar panels (155 W and 55W respectively) mounted on a pylon. Data are collected daily at 2 pm (CEST) and is transmitted via a modem and an omnidirectional antenna mounted in the wall of the



80 concrete chamber. For the configuration of the quadrupoles, we use a dipole-dipole skip two array, collecting 1'494 direct and reciprocal data points per day.

### 2.4 Data processing and analysis

Data shown here were processed and analysed for one day in summer (20 August 2021; close to the expected timing of the

85 maximum thickness of the active layer, generally around the end of August) and for one day six months later in mid-winter (20 February 2022).

To invert the measured apparent resistivities to specific resistivities of the near-subsurface, the Python-based open-source software ResIPy was used (Blanchy et al., 2020). The inverse modelling is based on Occam's principle. It is performed to minimize an objective function, which quantifies the misfit between the observed dataset and the predictions made by the

90 model (Binley and Slater 2020). The two presented datasets were inverted independently. The reciprocal data acquisition allows to estimate the data quality, filtering the ERT data, and hence, to eliminate implausible values and outliers. The direct and reciprocal deviations were checked (Blanchy et al., 2020), and only data with less than 5% discrepancy were considered for further inversion. Data error analyses were performed following Blanchy et al. (2020). The field data's error check was used to weight the later error model parameters in the inversion process. Since the current flow does not interact with the

95 surface or other boundaries, all electrodes except the top two were treated as buried in the modelling process and an unstructured triangular mesh was created, extending the real mesh laterally and downwards. The computed models converged after six (20 August 2021; RMS error = 1.06) and five iterations (20 February 2022; RMS error = 1.04), respectively. Both independent models clearly show the resistivity variation between the summer and winter datasets. However, to better highlight the variations, differences were additionally modelled using ResIPy`s time-lapse inversion scheme (Blanchy et al., 2020).

100 Model results were plotted with the open-source visualization application ParaView (paraview.org).

### 3 First results

The stratigraphies recorded indicate a layer of large boulders dominating the topmost 3-4 m of the three 2020 boreholes B3-B5. Below these, the substrate in B3 constituted of icy sediments and dirty ice (Fig. 1b). Some icy sediments were also found in B4 and B5, but wet sludge containing ice particles dominated. Ice and water distribution are thus heterogeneous over

105 distances of 5-10 m.

The borehole temperatures, piezometer data and cross-borehole ERT data shown for 20 August 2021 and 20 February 2022 highlight thermal and phase contrasts between summer and winter conditions (Fig. 3). Vertical temperature distributions in B1 (drilled in 1990) and SB5 are in good agreement and show that permafrost temperatures are close to 0°C in both seasons (Fig. 1a). The active layer is 3.5 – 4 m thick in both boreholes and active layer temperatures are nearly identical in August (Fig. 3a).

110 In February active layer temperatures are higher in B1 than B5, suggesting a higher moisture content in B1.

In the blocky active layer, the piezometer data mainly corresponds to air pressure. Below the active layer the presence of ice makes the quality of the data uncertain. Nevertheless, the pressure data at 6-10 m depth seems most plausible, as the substrate consists of wet sludge (with a high unfrozen water content) containing ice crystals. Pressures were lower in winter than in summer in this layer.

115 On 20 August 2021, the ERT image is consistent with the stratigraphies recorded in B3 and B4 in August 2020 and with the temperature data (Fig. 3b). Resistivities below 15 kΩm are mainly found in the unfrozen active layer. The blocky, uppermost layer can best explain the anomaly of higher resistivities (0-1.5 m depth, right). Resistivities are highest where ice was recorded in the stratigraphy, reaching values above 60 kΩm and indicating a lower water content. The relatively low resistivities between 7-9 m depth are remarkable. They reveal the stratigraphic contrast between the very ice-rich layers and the sludgy

120 layers containing ice crystals, with high water contents. In winter, resistivities increase by up to 350 % due to the general





relation between resistivity and temperature (Figs. 3b & c). The most remarkable resistivity changes occur in the active layer and the underlying sediments with ice (B3). In contrast, the resistivity changes are smallest near B4 due to the higher water content in the sludge-ice layers.

## 4 Discussion

The multi-method approach presented is a novel combination of techniques in an ice-rich rock glacier and has delivered first promising results. The temperature data measured in boreholes B1 and B5 are almost identical in summer but contrast strongly in winter in the active layer, highlighting local differences in moisture content. Below 4.5 m depth, ground temperatures are close to 0°C all year round, making it difficult to distinguish variations in ice-/water content thermally. The continuous cross-borehole ERT data thus provide useful supplementary information: they show vertical and lateral changes in resistivity,

allowing to discern phase changes in the active layer and variations in ice-/water content in the underlying permafrost. The quality of the contact between the electrodes and the sand-gravel borehole infill is unknown, but summer resistivities confirm summer borehole stratigraphies. This is encouraging, as the latter were determined from observations during destructive drilling and not on the basis of borehole cores. Little seasonal change in resistivity is registered in the wet sludge layers containing ice crystals, which is likely due to latent heat effects. These layers confirm that mountain permafrost can contain

substantial unfrozen water contents. The lower water contents in B3 (ice and sediments with ice) allow more efficient freezing in winter. The highest seasonal electrical resistivity changes are observed in ice-rich sediments.

The piezometer data must be interpreted with care: if ground temperature drops below 0°C, ice formation might strongly affect the pressure measured in the sensor's housing and thereby not fully represent the dominant pressure condition at a given depth. Laboratory experiments will be necessary to determine the behavior of these types of piezometer sensors, depending on ground

ice and water contents. Nevertheless, the piezometers do indicate the presence of air, water or ice and they register seasonal pressure variations in wet layers. All of the data presented here highlight the heterogeneous and seasonally variable nature of the substrate, as was revealed by the contrasting borehole stratigraphies. Frequency domain electromagnetometer data from 2019 (Boaga et al., 2020) additionally confirm this.

The main technical challenges confronted were: 1) borehole walls collapsing between the extraction of the drill head and

insertion of the sensors (requiring the use of a stabilizing PVC tube), and 2) establishing efficient contact between the sensors and the borehole walls. The filling material differed from the original substrate and it is not known whether it subsequently settled and whether all spaces around the sensors were efficiently filled. Processes such as subsidence, creep, and changing ice-/ water contents could be a challenge for long-term sensor health in these tubeless boreholes.

## 5 Conclusions

A novel combination of measurement techniques has been used to investigate ice and water contents in an ice-rich rock glacier in the Swiss Alps. Borehole piezometer measurements and cross-borehole electrical resistivity soundings give a first insight of seasonal changes in ice and water content and their thermal regime. Borehole stratigraphies in three 12 m boreholes indicate very heterogeneous substrate conditions within lateral distances of 5-10 m. Permafrost temperatures are close to 0°C. The

recently installed and unique combination of sensors reveals spatial and temporal variations in ice and water content, which cannot be discerned on the basis of borehole temperatures alone. The methods presented here provide a first valuable insight on local rock glacier substrate characteristics and relative ice-to-water ratios. Mid- to long-term monitoring will contribute towards understanding climate-related factors driving rock glacier kinematics and future water availability from ice-rich rock glaciers.






**Figure 1: a) Permafrost distribution at the Schafberg study site (Permafrost and ground ice distribution map of Switzerland, Kenner et al. 2019). Aerial photograph: swisstopo; b) Borehole stratigraphy in B3, B4 and B5 (drilled in 2020) and positions of the ERT electrodes (purple dots) and borehole piezometers/temperature sensors (yellow dots (B5) and pink dots (B1)). No stratigraphy is shown for B1 (drilled in 1990); b) Positions of the boreholes and instruments relative to each other and their position on the Schafberg rock glacier (1a, black rectangle).**


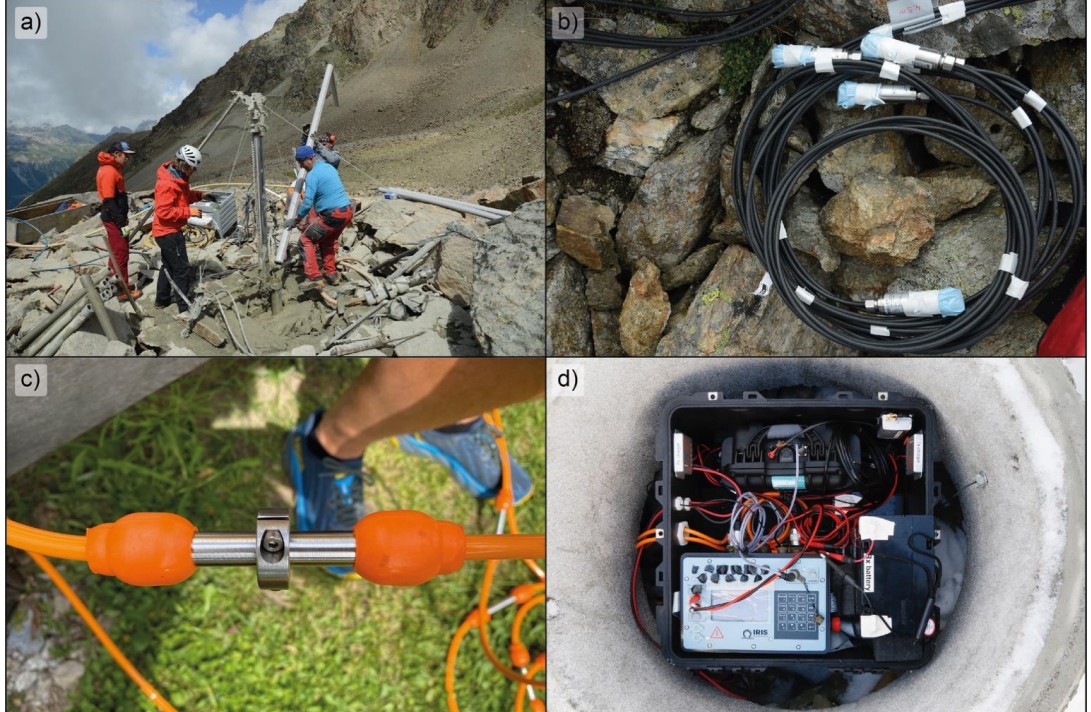

**Figure 2: a) Drill and protective PVC pipe used to drill boreholes B3, B4 and B5 on Schafberg rock glacier in August 2020; b) Piezometers and temperature sensors ready to be installed in B5; c) Stainless steel ring mounted on cross-borehole ERT electrode to improve ground contact; d) Cross-borehole ERT logging system for B3 and B4 in a concrete chamber (ERT instruments in Fig. 1c).**









**Figure 3: a) Left: Borehole temperatures in B1 (squares) and B5 (dots) on 20 August 2021 (yellow) and 20 February 2022 (blue), between 0 and 9 m depth. Right: Piezometer pressure data measured in B5 on 20 August 2021 (yellow) and 20 February 2022 (blue). The stratigraphy of B5 is shown in the background. b) Cross-borehole resistivities on 20 August 2021 (left; 6 iterations, RMSE 1.06) and on 20 February 2022 (centre; 5 iterations, RMSE 1.04). The stratigraphies of B4 and B3 are shown beside the 20 August 2021**

**resistivity profiles to underline the consistency between stratigraphies and resistivities (for legend see Fig. 3a). c) Percentage change of resistivities between 20 August 2021 and 20 February 2022. The black-dashed line indicates the 0% contour line.**





*Author contributions*. MP initiated the study, designed the measurement concept, drilled and instrumented the boreholes. CB carried out the electronics work and programmed the cross-borehole ERT with AB. AB, SW, MP, JB and MPa analysed the borehole data. All authors contributed to the manuscript.

*Acknowledgements*. Nora Bühler, Samuel Halter, Thomas Schlatter and Lars Widmer are thanked for their practical assistance. We thank the *Foffa Conrad AG* drilling team and *Helibernina AG* for their excellent work and the *Amt für Wald und Naturgefahren Graubünden* for their logistic support.

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
