# Peer review of "Brief communication: Combining borehole temperature, borehole piezometer and cross-borehole electrical resistivity tomography measurements to investigate seasonal changes in ice-rich mountain permafrost"

_The Cryosphere, 2022_

## Referee Comment (RC2)

[referee-annotated manuscript omitted]

---

## Author Comment (AC1)

**Authors' Comments AC_tc-2022-165**

Brief communication: Combining borehole temperature, borehole piezometer and cross-borehole electrical resistivity tomography measurements to investigate changes in ice-rich mountain permafrost (Phillips et al.)

**Reply to Referee #1**

*Dear Referee 1,*

*thank you very much for your positive and constructive feedback. Please find our replies to your suggestions below (in blue, **suggested changes in bold script**).*

*With kind regards,*

*Marcia Phillips, Chasper Buchli, Samuel Weber, Jacopo Boaga, Mirko Pavoni and Alexander Bast.*

L20: Consider one or two sentences that briefly summarize the major advances related to modern quantitative measurements of geophysical parameters on rock glaciers in the Alps, from e.g. Vonder Mühll and Haeberli (1990) until today.

*We suggest adding the following to the Introduction (Line 29 onwards):*

Many Alpine rock glaciers are close to their melting point (PERMOS 2019), and borehole temperature data do not allow to distinguish between ice and water close to 0°C, so relative changes in ice-/water content have to be monitored using geophysical methods detecting changes in resistivity (Mollaret et al., 2019). **Electrical resistivity methods have been established in mountain permafrost terrain in the past few decades and deliver increasingly detailed information, including quantitative estimation of water storage, ice contents, water flow and temperature (Hauck 2013).** Continuous surface electrical resistivity data (ERT) provide valuable 2D information on resistivity changes in ice-rich permafrost substrates.

(*We will thus add the following reference):*

**Hauck, C.: New Concepts in Geophysical Surveying and Data Interpretation for Permafrost Terrain, Permafrost and Periglacial Processes, 24, 131-137, https://doi.org/10.1002/ppp.1774, 2013.**

L23-27: Maybe worth mentioning one or some previous studies on rock glaciers that used similar approach as your study, with a combined application of geophysical techniques with cross-hole experiments (georadar cross-hole tomography). e.g. Maurer et al. 2003., Musil et al. 2006 and Springmann et al. 2012. Although these studies aimed at delineating internal structure and investigating the stability of rock glaciers, there are some relevant findings that could be discussed in light of your results.

*Thanks for the useful literature! We propose to modify the introduction thus:*

*Line 27 onwards:*

…However, there is little direct information on the internal hydrology of rock glaciers (Zenklusen Mutter and Phillips, 2012), changes in ice-/water contents or talik formation. A substantial unfrozen water content can persist well below 0°C, depending on soil properties,

salinity and pore water pressure (Arenson et al., 2022), **as was shown by Musil et al. (2006), using crosshole georadar measurements in the Muragl rock glacier.**

*Line 33 onwards:*

However, 2D ERT soundings can only provide limited information with increasing depth. **Cross-borehole measurements such as georadar (Musil et al 2006) or cross-borehole ERT overcome this limitation by using at least two vertical boreholes for instrument locations (Binley, 2015).** Until present, **cross-borehole ERT** has primarily focused on groundwater research, especially on the remediation of contaminated groundwater (Binley and Slater, 2020) or to determine substrate characteristics, but has never been applied in mountain permafrost environments.

L44: The use of "continues measurements" in this respect is somewhat misleading as data presented are only for one day in summer and for one day in mid-winter. Consider rephrase.

*To clarify we suggest the following change (please note: the measurements are designed to be carried out continuously and are running, but we only show two data snapshots in this Brief Communication):*

'Here we present **first data snapshots from** a novel combination of borehole temperature, borehole piezometer and cross-borehole ERT measurements designed to investigate the changes occurring in an ice-rich rock glacier, in particular modifications of ice-to-water ratio near 0°C, with continuous measurements'.

L137-138: Is there a citation to backup the statement regarding piezometer data and how ice formation might affect the pressure?

*To our knowledge, this is a first-time application of piezometers in ice-rich permafrost terrain. However, Harris & Davies (1998) made the following observation regarding their laboratory experiments:*

'In the present case it is argued that the rapid transition from negative to positive readings during freezing is in response to the sealing of the pressure transducer within an effectively closed frozen soil system. The transducer became isolated from pore fluid films and dominated by the positive ice pressures which develop during heave. Since there is considerable uncertainty as to the mechanism of pressure transfer from the heaving soil to the transducer it cannot be assumed that pressure readings necessarily accurately reflect ice pressures'.

*We therefore propose to add:*

The piezometer data must be interpreted with care: if ground temperature drops below 0°C, ice formation might strongly affect the pressure measured in the sensor's housing and thereby not fully represent the dominant pressure condition at a given depth. **Similar challenges were encountered by Harris and Davies (1998) in laboratory experiments. Further laboratory experiments will be necessary…**

L157-159: Include a couple of sentences with some more details about the future plans for this work and monitoring. What could a more in-depth analysis, and e.g. more data from all seasons and inter-annual variability add to new knowledge?

*To address future work and the associated increase in knowledge, we propose to delete the last sentence and replace it with a new paragraph:*

**Nevertheless, future analysis will reveal daily but also medium- to long-term interannual and inter-seasonal changes in rock glacier water content, which will be correlated with meteorological variables. This information contributes towards i) closing the gap regarding the direct quantification of rock glacier water content and ii) a better understanding of climate change impacts. The unique combination of methods presented here will provide valuable insight on local rock glacier substrate characteristics and relative ice-to-water ratios, thus contributing to understanding factors driving accelerating rock glacier kinematics and future water availability from these landforms.**

Literature suggested by Reviewer 1: *(we propose to cite Musil et al. (2006) in the Introduction)*

H.R. Maurer, S.M. Springman, L.U. Arenson, M. Musil, D. Vonder Mühll Characterisation of potentially unstable mountain permafrost — a multidisciplinary approach. Proceedings of the 8th International Conference on Permafrost, Zurich, Switzerland (2003), pp. 741-746

*To be added to the references:*

**Musil, M., Maurer, H.R., Hollinger, H. and Green, A.G., 2006. Internal structure of an alpine rock glacier based on cross hole georadar travel times and amplitudes. Geophysical Prospecting, 54 (3), 273–285.**

Springman, S.M., Arenson, L.U., Yamamoto, Y., Maurer, H., Kos, A., Buchli, T. andDerungs, G., 2012. Multidisciplinary investigations on three rock glaciers in the Swiss Alps: legacies and future perspectives. Geographiska Annaler: Series A, Physical Geography, 94, 215–243. doi:10.1111/j.1468- 0459.2012.00464.x

Vonder Mühll, D. and Haeberli, W., 1990. Thermal characteristics of the permafrost within an active rock glacier (Murtèl-Corvatsch, Grisons, Swiss Alps). Journal of Glaciology, 34 (123), 151–158.

---

## Author Comment (AC2)

**Authors' Comments AC_tc-2022-165**

Brief communication: Combining borehole temperature, borehole piezometer and cross-borehole electrical resistivity tomography measurements to investigate changes in ice-rich mountain permafrost (Phillips et al.)

**Reply to Referee #2**

*Dear Referee 2,*

*thank you very much for your positive and constructive feedback. Please find our replies to your suggestions below (in blue, **modifications in bold script**).*

*With kind regards,*

*Marcia Phillips, Chasper Buchli, Samuel Weber, Jacopo Boaga, Mirko Pavoni and Alexander Bast.*

Comments from pdf document:

P1:

Title : Even if the overall aim is not only seasonal change, the topic of this specific paper is seasonal changes. Inserted Text: seasonal

*We will change the title to:*

Brief communication: Combining borehole temperature, borehole piezometer and cross-borehole electrical resistivity tomography measurements to investigate **seasonal** changes in ice-rich mountain permafrost

P2:

Line 51: Please define PERMOS when it appears in the text for the first time.

*We will add*: …where an existing PERMOS **(Permafrost Monitoring Switzerland)** borehole…

Line 51: how deep?

*Add*: …(Schafberg B1, **drilled to 67m depth** in 1990; **lowest temperature sensor currently at 15.9 m**)

P3:

Line 92: It would be interesting to know how much data are actually filtered out (how many for both measurements?)

*We will change to*: … 'The direct and reciprocal deviations were checked (Blanchy et al., 2020), and only data with less than 5% discrepancy were considered for further inversion **(remaining data points: $n_{Aug21}$ = 214; $n_{Feb22}$ = 118)'.**

Line 96: What convergence criteria did you use?

*We will add*:

**The iteration process was terminated when it reached a maximum of ten iterations, or the data-model misfit based on the least-square fit equalled the number of measurements (Binley & Slater 2020).**

Line 99: I understand here that you performed individual independent inversions and in addition a time-lapse inversion. In that case, it would be interesting to see the comparison of the change in resisitivity derived from individual inversion and from the time-lapse inversion (in an Annex for example).

*We agree! However, this is something we are doing in a longer paper (Bast et al. in prep.) with more detailed analyses of the time series and which cannot be done in a Brief Communication.*

Line 109 x 2: the term 'thaw layer' may be more adequate than 'active layer' (as it is not sure that the measurement on August 20th was undertaken exactly when the active layer is completely thawed, right?

Otherwise please explicit that you based on more data than 2 datasets (as one cannot determine the active layer thickness with only 2 datasets).

*The active layer refers to the uppermost part of the ground which thaws and refreezes on a seasonal basis. Maximum active layer thickness is the maximum depth to which the 0°C isotherm penetrates in summer/autumn.*

*We suggest changing this to: … **the maximum active layer thickness is generally** 3.5-4.0 m in both boreholes and **temperatures in the thawed part of the active layer** are nearly identical in August…*

Line 110: This sentence can be misleading, as temperatures are indeed higher in B1 than B5 within the active layer, but also within the permafrost.

*Change to: **February temperatures are higher in B1 than B5,** suggesting a higher moisture content in B1.*

Line 113: remove *unfrozen* (from unfrozen water content), add *liquid* water content.

*Liquid water can exist at subzero temperatures in permafrost and is termed* unfrozen *water (see e.g.: Glossary of permafrost and related ground-ice terms, Harris, S. A.; French, H. M.; Heginbottom, J. A.; Johnston, G. H.; Ladanyi, B.; Sego, D. C.; van Everdingen, R. O., NRC Publications Archive, or various publications, e.g.: https://doi.org/10.1029/WR023i012p02279 or https://doi.org/10.1002/ppp.2031 or: The Periglacial Environment, p. 130 (Hugh French, 1996, Longman Ltd.). The use of the term implies that the ground temperature is below 0°C, but the water is still liquid – whereas the use of 'liquid water content' does not necessarily imply that the ground is frozen.*

*We therefore propose to keep the term **unfrozen water content**.*

Line 113: Here it would be interesting for the reader to better understand the temporal changes (or to know whether the change between summer and winter is continuous or if many short-term peaks occur). Can the higher summer pressure be related to water circulation?

*We will show the pore-pressure time series in a longer paper (Bast et al. in prep.). They change gradually, with maxima during snow melt infiltration.*

*We suggest: Pressures were lower in winter than in summer in this layer **(Fig. 3a), due to higher water contents in summer (infiltration of snow melt water and precipitation).***

Line 113: add somewhere in this paragraph the reference to Fig. 3a (i.e. to the pressure data).

*See suggestion above.*

Line 116: Thaw layer (maybe it is a better term, as 'below <0°C' does not mean frozen. The active layer in the case of a very porous material may actually be very dry).

*We suggest: '**in the thawed active layer**…'*

Line 118: Add liquid (liquid water content).

*We suggest keeping water content, as water is per se liquid (as opposed to the words used for it in other phases: ice/vapour, where one doesn't refer to solid water/gaseous water).*

Line 120: the logarithmic change needs to be used, as logarithmic scales can much better than a linear scale represent the high variations of electrical resistivity. The logarithmic scale is widely used in the permafrost community, see for instance your own references about electrical resistivity: Permos, 2019 or Mollaret et al. 2019.

*Both logarithmic and linear scales are frequently used in the (permafrost) literature. We deliberately chose the linear scale with absolute values to highlight the differences between the summer and the winter season. We feel the minor differences in the "icy sludge" (left) area become more evident than they would with a logarithmic scale. Furthermore, we only compare two data sets/time points and not several periods, as, for example, in Mollaret et al. 2019, where a logarithmic scale can provide a better insight.*

Line 120: liquid (or higher liquid water contents) and Line 123

*See response for Line 118.*

Line 131: Did you or can you measure the contact resistance at each electrode

*Unfortunately, we do not measure the contact resistance. We discussed this when setting up the experiment since it is an excellent idea. It is certainly possible to measure the contact resistance with every sounding. However, due to energy-related reasons, it is not feasible. We operate the system with a solar panel in a harsh alpine environment. The power is enough for one daily measurement and usually for daily data transmission. In the winter season, we already reach the limits of the power supply. We therefore decided not to risk data loss related to the energy supply and thus to accept not monitoring the contact resistances.*

Line 145: add *electrical*

*We agree: (establishing **electrical** contact between the sensors…).*

Line 146: It will also be interesting to see in the future if the contact resistance changes over time (I mean not the seasonal change but annual/long-term), to see for instance is part of the filling material drained away. This is why it may be important to monitor the contact resistance at each electrode.

*We agree. However, please see our comment for L131.*

*Furthermore, we already stated in the manuscript (from L130 on): 'The quality of the contact between the electrodes and the sand-gravel borehole infill is unknown, but summer resistivities confirm summer borehole stratigraphies'.*

Line 146: what did you use as filling material?

*Please see Line 55 onwards*:

'The sensors were lowered into the boreholes immediately after drill extraction, the stabilizing PVC pipe removed and the boreholes filled with a sand/gravel mixture (total volume 0.3 m$^3$, consisting of 50% sand (≤2 mm diameter) and of 50% gravel (2-4 mm diameter)) to establish contact between the sensors and borehole walls (see sections 2.2 and 2.3) and to minimize air circulation.

Line 157: Looking forward to seeing more on this!

*Thank you, we are going to publish a more detailed paper (Bast et al. in prep.) and are also excited to see how things will develop!*

Line 166 (figure legend): model or modelled permafrost distribution

*Change to*: Figure 1: a) **Modelled permafrost** distribution

Figure 3a: The choice of the stratigraphy legend style of the 'coarse sediments with ice' may be misleading (i.e. between 3.5 and 4.5 m depth. Here you assume that the active layer is of 3.5 m, and it seems that the coarse sediments are saturated with ice. However, from the pressure data (and also as you explain in the rest of the paper), the liquid water and also the air content (from the pressure data) are also not negligible. Please try to re-word to make it more explicit that the layer between 3.5 and 4.5 m is most probably not saturated with ice.

*We assume the active layer is 3.5 m thick and that the underlying ground consists of coarse sediments with ice in B5, based on the information derived from the drilling stratigraphy and the borehole temperature data. The piezometer data are to be considered carefully, as we stated in Line 137: 'The piezometer data must be interpreted with care: if ground temperature drops below 0°C, ice formation might strongly affect the pressure measured in the sensor's housing and thereby not fully represent the dominant pressure condition at a given depth'.*

*So we feel the legend style is correct here.*

Legend Figure 3a: In this plot the ice legend is not used, right? So I suggest to remove it from the legend.

*That is correct, and we can remove the pure ice symbol from the legend (Fig. 3a). However, since the stratigraphy of the two boreholes (B3 and B4) is plotted next to the tomograms in Fig. 3b, we prefer to leave the symbol in the legend and keep our reference to Fig. 3a in the figure caption (see L185).*

Figure 3b: As already mentioned, a logarithmic scale would be more appropriate. Same of the change in resistivity. I would recommend to use the change in the resistivity logarithm (used both in Permos 2019 and Mollaret et al. 2019).

*Please see the response for L120.*

Line 186, Figure legend 3c: It would be good to mention that there is part of the tomogram where the resistivity seems to have decreased and to discuss why. What are the measurement uncertainties? Is the resistivity decrease for real or due to any artifact from your point of view? In case you are not sure you should explicit it.

*Indeed, resistivities seem to decrease from summer to winter in the tomogram's left part (B4). We are convinced that the winter resistivity profile clearly shows the presence of icy sludge in B4 with low resistivities (left side), whereas B3 (right side) has higher resistivities in winter. The lower water contents in B3 allow more efficient freezing in winter, thus explaining the*

*resistivity models. However, we feel that it is too early to deliver a final answer based on the two presented resistivity models. Further geophysical modelling is crucial and will be carried out using data with a higher temporal resolution. Based on the available data for this TC Brief communication, we cannot make any statements on uncertainties. However, as described in the manuscript, the summer model strongly agrees with the recorded stratigraphies.*

---

## Author Response (AR1)

Marcia Phillips
WSL Institute for Snow and Avalanche Research SLF
Flüelastrasse 11
7260 Davos Dorf
Switzerland

phillips@slf.ch, +41 81 417 02 18

Response to Editor, 11th January 2023

**Brief Communication TC-2022-165, submitted by Phillips et al.**

Dear Professor Booth,

please find attached our manuscript with our suggested changes (one version with the track changes modifications visible, one in which the modifications have been accepted).

We hope the revised manuscript will meet your approval and look forward to hearing from you.

With best wishes for the new year and kind regards,

Marcia Phillips and co-Authors